# Secure Encapsulation Schemes Using Key Recovery System in IoMT Environments

**DOI:** 10.3390/s21103474

**Published:** 2021-05-17

**Authors:** Taehoon Kim, Wonbin Kim, Daehee Seo, Imyeong Lee

**Affiliations:** 1Department of Software Convergence, Soonchunhyang University, Asan 31538, Korea; 20134101@sch.ac.kr (T.K.); wbkim29@sch.ac.kr (W.K.); 2Faculty of Artificial Intelligence and Data Engineering, Sangmyung University, Seoul 03016, Korea; daehseo@smu.ac.kr

**Keywords:** CL-PKC, key encapsulation, key recovery system, proxy re-encryption, signcryption

## Abstract

Recently, as Internet of Things systems have been introduced to facilitate diagnosis and treatment in healthcare and medical environments, there are many issues concerning threats to these systems’ security. For instance, if a key used for encryption is lost or corrupted, then ciphertexts produced with this key cannot be decrypted any more. Hence, this paper presents two schemes for key recovery systems that can recover the lost or the corrupted keys of an Internet of Medical Things. In our proposal, when the key used for the ciphertext is needed, this key is obtained from a Key Recovery Field present in the cyphertext. Thus, the recovered key will allow decrypting the ciphertext. However, there are threats to this proposal, including the case of the Key Recovery Field being forged or altered by a malicious user and the possibility of collusion among participating entities (Medical Institution, Key Recovery Auditor, and Key Recovery Center) which can interpret the Key Recovery Field and abuse their authority to gain access to the data. To prevent these threats, two schemes are proposed. The first one enhances the security of a multi-agent key recovery system by providing the Key Recovery Field with efficient integrity and non-repudiation functions, and the second one provides a proxy re-encryption function resistant to collusion attacks against the key recovery system.

## 1. Introduction

In the era of the Fourth Industrial Revolution, as various countries and companies around the world have heavily invested in Information Technology (IT), the emergence of Internet of Things (IoT) environments has increasingly enabled a convenient and broad diversity of services to be distributed to consumers via various types of smart devices. There are various systems such as the Internet of Medical Things (IoMT), Intelligent Transportation Systems (ITS), smart home appliances, and connected cars that have been implemented on those smart devices and deploy a vast number of services to consumers [1,2]. Therefore, many current types of research have applied those IoT technologies to various environments.

Although the development of IoT has increased device convenience, it has also been accompanied by increasing threats to national, corporate, and personal information security [3]. According to the security threats, such as personal information leakage cases, encryption has rapidly become important to secure personal information [4]. Therefore, the importance of security issues in IoT environments has also increased. Furthermore, there is discussion regarding security issues related to key management, in which problems may arise where ciphertexts cannot be decrypted if the keys are lost or corrupted.

In general, key recovery is a system that provides the ability to reveal the key to an authorized user under specific conditions specified in advance [5]. This paper presents schemes to recover lost or corrupted keys using an encapsulation-based key recovery system. When a user needs a key that was used to create a ciphertext, a newly defined field known as Key Recovery Field (KRF) can be used to recover the key. If key is lost or corrupted, the recovered key can be used to decrypt the ciphertext. Because security is necessary for key management and recovery in various environments using IoT, there is much research on key recovery systems for use with IoT. Guo et al. [6] proposed a secure group key distribution scheme for untrusted wireless networks. Guo et al. used the Self-healing Group Key Distribution (SGKD) protocol to ensure group communication security and improve communication efficiency. Instead of requiring the group manager to resend the missing key to update the message, Guo et al. proposed a scheme for group members to recover the lost session key from the current broadcast message.

Lee et al. [7] proposed an efficient and secure key distribution and key recovery mechanism suitable for the characteristics of the IoT environment. The proposed system added the key recovery function required to prevent the reverse function of the encryption and key recovery, providing security to both due to the communication device could not unilaterally recover the key. In addition, it is efficient because there is little information sent during key recovery, which is suitable for IoT environments.

Sung [8] proposed a scheme to support secure sensor data for cloud computing to activate services at the IoT application level. Sung proposed key management that enables continuous key authentication for the privacy of sensing information in such a cloud computing environment and enables secure recovery if the key is lost or corrupted.

Losing a key in an IoMT environment will prevent access to information such as previous medical treatment data and information on medications being taken, and impede accurate medical examination and treatment. Therefore, key management is important in IoMT environments [9]. There are four agents in a key recovery system in IoMT. The key Generation Center (KGC) can generate some parameters of a network participant’s public key pair. The Medical Institutions (Med) have all the medical treatment data, The key Recovery Center (KRC) can recover the complete key. The key Recovery Agent or Key Recovery Auditor (KRA) can share the KRC’s key recovery operations or monitor other agencies. If the patient loses the key used for the ciphertext, the key can be recovered with the help of remains the KRF, Med, KRC, and KRA.

However, the problem remains that the KRF may be forged or altered by a malicious user. To solve this problem, we propose our first scheme, which efficiently provides integrity and non-repudiation functions for the KRF and enhances the security of a multi-agents key recovery system.

The main contributions of our proposed scheme-I as follows:It provided a key recovery system based on secure encapsulation against various types of attacks and provides the ability to securely recover a lost or corrupted key.It uses signcryption to ensure KRF integrity and non-repudiation. In addition, it provides both digital signing and encryption at the same time to increase computational efficiency.It uses values that only authorized KRAs hold to prevent unauthorized KRAs and group-based authentication attacks. If some KRAs do not perform the key recovery properly, key recovery may be performed by other authenticated KRAs to prevent a single point of failure.

Furthermore, the Med, KRA, and KRC may collude and behave maliciously. To solve this problem, we propose scheme 2, which provides a proxy re-encryption function and enhances the security of a key recovery system against various types of attacks such as collusion attacks and the key escrow problem.

The main contributions of our proposed scheme-II as follows:It prevents the Med, KRC, and KRA from behaving maliciously to recover keys without authorization and prevents unauthorized entities from obtaining keys.It uses a partial private key generation scheme to prevent the KGC from generating private keys for all participants.

The remaining parts of the paper are organized as follows. Section 2 describes related work, and Section 3 describes system model for the proposed schemes. Section 4 describes scenarios and detailed protocols for the proposed scheme-I, and Section 5 describes scenarios and detailed protocols for the proposed scheme-II. Section 6 analyzes whether the proposed schemes satisfy the security requirements. Finally, Section 7 discusses our conclusions.

## 2. Related Work

This section reviews and discusses existing works related to key recovery systems and encryption schemes.

### 2.1. Encapsulation Key Recovery Systems

A key recovery system is an important part of an encryption system. If a private key or session key used for a ciphertext is lost or corrupted, or a Law Enforcement Agency (LEA) wishes to intercept suspicious ciphertexts lawfully, it must be possible to recover the key. There have been several proposals related to such key recovery systems. Kanyamee et al. [10] proposed a highly available distributed session key recovery system. It provides high availability and attack detection for secure session key management and group authentication while using Multi-Key Recovery Agents (M-KRA) to solve the single point of failure problem encountered in the traditional KRA approach. However, many problems remain, such as the risks of forgery, counterfeiting, and collusion attacks for user-generated KRFs, which can cause problems for the key recovery service.

Lim et al. [11] proposed an encapsulation-based M-KRA key recovery system. They attempted to solve the problem that the M-KRA must communicate directly with one or more KRAs in existing M-KRA scheme, and the user must directly perform a complex key recovery process. Their scheme provides secure session key management and recovery using a new type of M-KRA to solve this problem. However, problems may arise in the key recovery service the forgery or modification of KRFs and non-repudiation problems related to user-generated KRFs.

Kyusuk et al. [12] proposed an identity-based key escrow scheme to prevent malicious key use by LEAs. If an LEA maliciously obtains the key, it can read the encrypted data to the desired user. In other words, an LEA can intercept and obtain the users’ keys to read all encrypted data. To solve this problem, the scheme prevents LEAs from obtaining a key by themselves after generating a user’s key pair with the KGC generated master key and the user’s ID. However, since it is a single KRA, it is vulnerable to problems such as a single point of failure weakness and group authentication attacks, causing problems with the key recovery service.

Huadpaknam [13] proposed the Security Key Recovery System with Channel Quality Awareness (SKRS-CQA) for smart grid applications. If a Smart Meter Unit (SMU) loses the keys used for correcting to the smart grid, it needs to be recovered. To solve this problem, key recovery proposed, providing improved reliability, system availability, and data confidentiality. In addition, system reliability was improved by using amplification and forwarding relay protocols and a cooperative communication network with optimal power allocation.

### 2.2. Multi-Agent Key Recovery

A single agent key recovery system is associated with service overload and security problems. Therefore, we use a multi-agent (at least two agents) key recovery system. The multi-agents receive a ciphertext that contains a key from the user or the KRC. Later, KRAs send pieces of the key to the KRC to allow the KRC to recover the complete key. However, various attacks and security breaches are possible, and efforts have been made to deal with these issues [14]. In our key recovery system using signcryption, we security by increasing availability and enhance security.

### 2.3. Signcryption

Encryption and digital signatures are two encryption tools that can ensure confidentiality, integrity, and non-repudiation. Until 1997, cryptographic systems used separate components to provide these security functions. In public key schemes, the traditional scheme is to digitally sign the message and then perform encryption (signature-then-encryption). However, there are two problems: the operation efficiency is low and the cost is high. To solve this signcryption was proposed In 1997 Zheng [15] proposed the first signcryption scheme. Signcryption simultaneously performs digital signature and encryption. Signcryption compared to the traditional signature-then-encryption scheme, can effectively improve computational efficiency, by reducing computational cost and communication overhead. In addition, many other signcryption schemes have been proposed throughout the years, each of them having its problems and limitations while offering different levels of security and computational cost [16,17].

### 2.4. Secret Sharing

Secret sharing schemes are ideal for sensitive information. These pieces of information should kept highly confidential, as their exposure could be disastrous. However, it is also critical that they should not be lost. Traditional encryption schemes are not suitable for achieving a high level of confidentiality and stability at the same time. When storing encryption keys, the user has to choose between keeping a single copy of the key in one location or multiple copies of the key in multiple locations for maximum security. The secret sharing scheme proposed by Shamir and Blakley [18,19] in 1979 is a scheme of dividing the secret value into several pieces so that the secret value can be recovered only when more than a certain number of pieces are collected. Such a scheme is called Shamir’s (k, n) threshold scheme. This scheme divides the secret value into n pieces and entities may recover the secret value only when more than k pieces are collected. In another type of secret sharing scheme, there is one dealer and n players. The dealer gives a share of the secret to the players, but only when specific conditions are fulfilled will the players reconstruct the secret from their shares. The dealer accomplishes this by giving each player a share so that any group of t (for threshold) or more players can together reconstruct the secret but no group of fewer than t players can. In addition, many other secret sharing schemes have been proposed throughout the years with as in the care of signcryption, each of them having its problems and limitations while offering different levels of security and computational costs [20,21].

### 2.5. Proxy Re-Encryption

A Proxy Re-Encryption (PRE) scheme is a scheme that converts the ciphertext so that a proxy server can decrypt the ciphertext encrypted with user A’s public key using user B’s private key. In 1998, Blaze et al. [22] proposed the first two-way proxy re-encryption scheme. This scheme was designed using the ElGamal encryption scheme [23]. In 2007, Green et al. [24] proposed an ID-based proxy re-encryption scheme using ID-based encryption for the first time to solve the certificate management problem of the existing Public Key Infrastructure (PKI) based proxy re-encryption. ID-based encryption is a scheme of using the user’s identity as a public key [25]. In this scheme, the user’s identity itself is owned, so unlike in PKI-based environments there is no need to issue and manage certificates. In addition, since the KGC generates a private key corresponding to the identities and issues them to the users, it has the advantage of performing verification of the user through KGC in case of a dispute. However, the KGC issues all users’ private keys, which causes a key escrow problem in which KGC knows the private keys. Therefore, to solve this problem, a Certificateless Public Key Cryptography (CL-PKC) system was developed. The CL-PKC scheme was proposed by Al-Riyami et al. [26], and it solves the key escrow problem by issuing partial private keys to the users by combining the user’s identity and a random number. Building on these feature, in 2010, Sur et al. [27] proposed Certificateless Proxy Re-Encryption (CL-PRE) using CL-PKC. CL-PRE is currently a representative form of secure PRE because it can perform the purpose of proxy re-encryption without suffering the PKI certificate management problem or IBE key escrow problem [28,29,30].

## 3. System Model

This section describes the system models, system objects, and security requirements of the proposed schemes.

### 3.1. Common Proposed Key Recovery System Model

In this section, we present the two key recovery system models proposed in this study. Before describing each proposed model, we present the common elements of the proposed models.

#### 3.1.1. Common Design Goals of Proposed Schemes

The two key recovery system models presented in this research were designed in different forms. However, the basic goal of both models is encapsulated key recovery. The first model proposed in this study is a key recovery system using signcryption. This process involves recovering the session key used for communication by using the encapsulated key recovery field. The second model proposed in this study is a key recovery system using proxy re-encryption. The basic goal is the same as the first model described above. However, the design and additional goals of the two models differ from each other, The similarities and differences between the two models can be seen in Figure 1, which will be described in detail below.

#### 3.1.2. Common Objects of Proposed Schemes

The composition of the two system models proposed in this study can be seen in Figure 1. In Figure 1, the difference between M-KRA and KRA methods is shown for the types of participants in the two models. The remaining differences are detailed in each model’s respective section.

Key Generation Center (KGC): Every participant Part must perform the KCG and key generation and communication steps to generate keys. All Part can generate a private key through the private key generation step with KGC, and a public key corresponding to the private key can be generated. The KGC publishes the public parameter params for performing encrypted communication with Part.Devices (Dev): Dev are medical devices and monitoring devices. Devices perform communication in the system managed by the Med. In this model, Devs must perform communication in the format designated by Med, and the basic format follows the form of (C‖KRF), in which the ciphertext and KRF are concatenated. Devices participating in the communication need Med’s public parameters in order to make the session key used for message encryption into KRF. Furthermore, the generated KRF should be designed to only be controlled by KRC and KRA.Medical Institution (Med): Med is a medical institution that manages device authorization control and data on medical devices. When a device requests KRF key recovery, the Med verifies that it is the lawful owner of the KRF. In this paper, the step of confirming whether the KRF is a lawful owner is omitted. In addition, the Med sends the KRF to KRC to help recover the key.

### 3.2. Proposed Scheme-I(Key Recovery System Using Signcryption)

This section describes additional elements of the key recovery system model using signcryption, excluding the common elements of the two models proposed in Section 3.1.

#### 3.2.1. Design Goals of Proposed Scheme-I

The model of the key recovery system using signcryption is a key recovery system that is used when a device key is lost or corrupted as shown in Figure 2. The device requests key recovery from Med and sends KRF. The Med receiving the KRF verifies that it is a lawful device of KRF. If it is a lawful device, it requests KRC to recover the key and sends KRF. After receiving KRF, KRA decrypts the KRF and sends the obtained KRF pieces to the M-KRA. Then, after receiving the pieces of KRF, M-KRA decrypts them and sends the session key pieces to KRC. It collects the session key pieces, generates a complete session key, and sends it to the device.

#### 3.2.2. Objects of Proposed Scheme-I

The system objects of the key recovery system using signcryption is shown in Figure 2. In addition, M-KRA additionally exists, and its roles are as follows:Participants: Part represents all participants (Dev,Med,KRC,M-KRA) who use the encrypted communication provided by KGC. Part can perform encrypted communication only by using params provided by KGC.Key Recovery Center (KRC): KRC is an organization in charge of key recovery and plays a central role in key recovery. The key recovery process is performed according to Med’s request for key recovery, and KRF is converted into a form that can be recovered using KRC’s private key. In this model, to reduce the burden of KRC’s key recovery operation, the help of *M*-KRA is needed.Multi-Key Recovery Agents (*M*-KRA): *M*-KRA is the agent that helps some operations of key recovery by reducing the burden on KRC. The KRA included in the *M*-KRA determines whether the KRF is suitable for recovery to prevent abuse of the KRF’s authority. When receiving a key recovery request from KRC, *M*-KRA perform the KRF recovery process using their private key. Furthermore, *M*-KRA send the obtained session key pieces to KRC.

#### 3.2.3. Security Requirements of Proposed Scheme-I

The security requirements of the key recovery system using signcryption are as follows:KRF integrity: No participant in key recovery can maliciously transform KRF information from the device and KRF information required for key recovery cannot be changed.Data confidentiality: It should be possible for only authorized devices to decrypt encrypted data.Non-repudiation: The device should not be able to reject the fact that it generated the KRF. In addition, the fact that device-generated KRF should be clear after transmission, exchange, communication, and processing.Attack on group authentication detection: If a malicious third-party KRA pretends to be a lawful member of the key recovery group, KRA should be detected through group verification.Single point of failure protection: In *M*-KRA, some KRAs should be able to recover session keys even if another KRA fails to operate properly.

### 3.3. Proposed Scheme-II (Key Recovery System Using Proxy Re-Encryption)

This section describes additional elements of the key recovery system model using proxy re-encryption, excluding the common elements of the two models proposed in Section 3.1.

#### 3.3.1. Design Goals of Proposed Scheme-II

The model of the key recovery system using proxy re-encryption is a key recovery system that is used when a device key is lost or corrupted as shown in Figure 3. The device requests key recovery from Med and sends KRF. The Med receiving the KRF verifies that it is a lawful device of KRF. If it is a lawful device, it generates a re-encryption key. Then, it requests key recovery from KRC and sends the obtained KRF and the re-encryption key. After receiving the KRF and re-encryption key, the KRA partially calculates KRF and sends the partially calculated KRF to KRA. After receiving the partial calculated KRF, KRA performs some calculations and sends partial calculated KRF to KRC. After receiving KRF, KRC sends it to the Med. The Med decrypts it, generates a session key, and sends it to the device.

#### 3.3.2. Objects of Proposed Scheme-II

The system objects of the key recovery system using proxy re-encryption is shown in Figure 3.

Participants (Part): Part represents all participants (Dev,Med,KRC,KRA) who use the encrypted communication provided by KGC. Part can perform encrypted communication only by using params provided by KGC.Key Recovery Center (KRC): KRC is an organization in charge of key recovery and plays a central role in key recovery. The key recovery process is performed according to Med’s request for key recovery, and KRF is converted into a form that can be recovered using KRC’s public key. However, in this model, key recovery can only be completed with the help of KRA to prevent abuse of privileges by KRC.Key Recovery Auditor (KRA): KRA is a monitoring agency that judges whether a key can be recovered by auditing the validity of key recovery. The KRA determines whether KRF is suitable for recovery to prevent abuse of authority through collusion between the Med and the KRC. If the key recovery request is deemed to be lawful, KRA will perform the KRF recovery process with its private key and sends it over to the KRC.

#### 3.3.3. Security Requirements of Proposed Scheme-II

The security requirements of the key recovery system using proxy re-encryption are as follows:KRF integrity: No participant in key recovery can maliciously transform KRF information from the device and KRF information required for key recovery cannot be changed.Data confidentiality: It should be possible for only authorized devices to decrypt encrypted data.Med applied for support: The session key used for communication must be encrypted and stored in KRF. In the event of an emergency when it is necessary to view the device’s data, the encrypted session key must be able to recover the encrypted message according to the procedure determined by Med as needed.Collusion attack resistance: Fewer than three participants among the Med, KRC, and KRA should not be allowed to obtain keys even if they are maliciously colluding.Key escrow problem: KGC can generate private keys for all participants, but the complete private key must not be known.

## 4. Proposed Scheme-I (Key Recovery System Using Signcryption)

In this section, we propose a key recovery scheme using signcryption. This scheme is a scheme for recovering the lost or corrupted device’s key. This is mainly composed of a setup phase, a key pair generation phase, a session key exchange and encryption phase, a KRF generation phase, and a session key recovery phase as shown in Figure 4.

### 4.1. System Parameters

The system parameters used in the proposed scheme-I are as follows.

***p*:** Prime number***q*:** Prime factor of *p*-1**G:** Cyclic group on prime *p****g*:** Random generator, g∈G***H*:** Hash function, ${\left\{0,1\right\}}^{*}\times G\to Z_{p}^{*}$**$sk_{M}$:** Master private key, $sk_{M}\in Z_{p}^{*}$**$pk_{M}$:** Master public key, $pk_{M}=g^{sk_{M}}$**$Dev_{A}$:** Monitoring devices**$Dev_{B}$:** Medical devices**$Part_{i}$:** Network Participant *i*, ($Dev_{A},Dev_{B},Med,KRC,KRA\in Part_{i}$)**$w_{i},t_{i},z_{i},v_{i}$:** Random numbers, $w_{i},t_{i},z_{i},v_{i}\in Z_{p}^{*}$**$sk_{i}$:**$Part_{i}$’s private key, $sk_{i}=\left(d_{i},z_{i},v_{i}\right)$**$pk_{i}$:**$Part_{i}$’s public key, $pk_{i}=\left(X_{i},Z_{i},V_{i}\right)$**a,b:** Secret value of $Dev_{A}$ and $Dev_{B}$, $a,b\in Z_{p}^{*}$**$PSK_{A},PSK_{B}$:** Partial session key of $Dev_{A}$ and $Dev_{B}$**SK:** Session key between $Dev_{A}$ and $Dev_{B}$***x*:** Random number, $x\in Z_{p}^{*}$ with x≠p−1**$R_{KRA_{i}}$:** Random number of $KRA_{i}$, $R_{KRA_{i}}\in Z_{p}^{*}$**SGN:** Group authentication values assigned to agents (Shared Group Number)**c,r,s:** Signcryption values**$c_{i},r_{i},s_{i}$:***i*th signcryption pieces**$TT_{i}$:** Value containing the value to be recovered when some KRAs fail the key recovery operation**$Tc_{i},Tr_{i},Ts_{i}$:***i*th $TT_{i}$ pieces**M:** Message space, $M\in {\left\{0,1\right\}}^{n}$***M*:** Plaintext message between $Dev_{A}$ and $Dev_{B}$ (M∈M)***C*:** Ciphertext message (Encrypted *M*)**KRF:** Key recovery field, $E_{pk_{KRC}}(KRF_{1}||KRF_{2}||\cdots ||KRF_{n}\left||H\left(SGN\right)\right)$**$KRF_{i}$:***i*th key recovery field piece, $E_{pk_{KRA_{i}}}(c_{i}\left|\right|r_{i}\left|\right|s_{i}||SGN||TT_{i})$

### 4.2. Setup Phase

In this phase, the KGC takes the security parameters as an input the security parameter $1^{\lambda }$ and generates public parameters.

**Step 1:** The KGC selects λ-bit large prime *p*, where *q* is a large prime factor of p−1 and group G of prime order *p*. In addition, a random generator g∈G is selected.**Step 2:** A master private key $sk_{M}\in Z_{p}^{*}$ is randomly selected and a master public key $pk_{M}\in g^{sk_{M}}$ is computed.**Step 3:** KGC selects Hash function *H*.**Step 4:** Then, public parameters params=(G,n,p,q,g,S,H) are published.

### 4.3. Key Pair Generation Phase

In this phase, $Part_{i}$ receives a partial private key from KGC and uses it to generate full private key $sk_{i}$ and public key $pk_{i}$.

**Step 1:** KGC generates parameters $w_{i},t_{i}\in Z_{p}^{*}$ for participant $Part_{i}$ through the following operation and sends them to $Part_{i}$ through a secure channel.
(1)$$\begin{array}{c} X_{i}=g^{x_{i}} \end{array}$$
(2)$$\begin{array}{c} d_{i}=x_{i}+sH\left(ID_{i},w_{i}\right)\;mod\;q \end{array}$$**Step 2:** Participant $Part_{i}$ who receives $X_{i},d_{i}$ from KGC, selects Random numbers $z_{i},v_{i}\in Z_{p}^{*}$ and sets $Part_{i}$’s private key $sk_{i}$.**Step 3:** Participant $Part_{i}$ generates $Z_{i},V_{i}$ and sets public key $pk_{i}$.
(3)$$\begin{array}{c} Z_{i}=g^{z_{i}} \end{array}$$
(4)$$\begin{array}{c} V_{i}=g^{v_{i}} \end{array}$$

### 4.4. Session Key Exchange and Encryption Phase

In this phase, the key recovery system uses signcryption to ensure integrity and non-repudiation and performs encryption of the session key simultaneously as shown in Figure 5.

**Step 1:**$Dev_{A}$ selects $a\in Z_{p}^{*}$ and calculate partial session key $PSK_{A}=g^{a}$. $Dev_{B}$ also selects $b\in Z_{p}^{*}$ and calculates partial session key $PSK_{B}=g^{b}$. After that, $Dev_{A}$ and $Dev_{B}$ exchange $PSK_{A}$ and $PSK_{B}$ with each other.**Step 2:**$Dev_{A}$ and $Dev_{B}$ calculate the session key $SK={\left(PSK_{B}\right)}^{a}={\left(PSK_{A}\right)}^{b}$ using the exchanged values $PSK_{A}$ and $PSK_{B}$.**Step 3:**$Dev_{A}$ generates random number $x\in Z_{p}^{*}$ and $k=pk_{KRC}^{x}\;mod\;p$, which is then divided in half into $k_{1}$ and $k_{2}$.
(5)$$k=(k_{1}\|k_{2})$$**Step 4:**$Dev_{A}$ generates c,r and *s* using $k_{1},k_{2},sk_{A},pk_{A}$ and SK.
(6)$$c=E_{k_{1}}\left(SK\right)$$
(7)$$r=H_{k_{2}}\left(g^{x}\;mod\;p,SK\right)$$
(8)$$s=x/(r+sk_{A})\;mod\;q$$**Step 5:**$Dev_{A}$ divides c,r and *s* to $c_{i},r_{i}$ and $s_{i}$.
(9)$$c=c_{1}⊕c_{2}⊕\ldots ⊕c_{n}$$
(10)$$r=r_{1}⊕r_{2}⊕\ldots ⊕r_{n}$$
(11)$$s=s_{1}⊕s_{2}⊕\ldots ⊕s_{n}$$
where *n* is the number of KRA and $c_{i}=\left\{c_{1},\cdots ,c_{n}\right\},r_{i}=\left\{r_{1},\cdots ,r_{n}\right\},s_{i}=\left\{s_{1},\cdots ,s_{n}\right\}$.

### 4.5. KRF Generation Phase

In this phase, when the key is lost or corrupted, the necessary KRF is generated to recover the key as shown in Figure 6.

**Step 1:**$Dev_{A}$ requests SGN to M-KRA.**Step 2:** Each of the KRAs requested for SGN from $Dev_{A}$ randomly selects $R_{KRA_{i}}\in Z_{p}^{*}$. After that, each KRA generates an SGN by sharing $R_{KRA_{i}}$ generated through a secure channel with each other.
(12)$$SGN=R_{KRA_{1}}⊕R_{KRA_{2}}⊕,\ldots ,⊕R_{KRA_{n}}$$**Step 3:** M-KRA send SGN to $Dev_{A}$.**Step 4:**$Dev_{A}$ generates $Tc_{i},Tr_{i},Ts_{i}$ using $c_{i},r_{i},s_{i}$ and SGN. Then, $TT_{i}$ is generated using $Tc_{i},Tr_{i}$, and $Ts_{i}$.
(13)$$Tc_{i}=c_{i}⊕SGN$$
(14)$$Tr_{i}=r_{i}⊕SGN$$
(15)$$Ts_{i}=s_{i}⊕SGN$$
(16)$$TT_{i}=(Tc_{i}\|Tr_{i}\left\|Ts_{i}\right)$$**Step 5:**$Dev_{A}$ generates KRF using $KRF_{i}$.
(17)$$\begin{array}{c} KRF_{i}=E_{pk_{KRA_{i}}}(c_{i}\|r_{i}\|s_{i}\|SGN\|TT_{i}) \end{array}$$
(18)$$\begin{array}{c} KRF=E_{pk_{KRC}}(KRF_{1}\|KRF_{2}\|\ldots \|KRF_{n}\left\|H\left(SGN\right)\right) \end{array}$$**Step 6:** Then, the generated KRF is attached to the ciphertext *C*.
(19)$$\begin{array}{c} \left(C\|KRF\right) \end{array}$$

### 4.6. KRA Fault Recovery Phase

In this phase, if some KRAs fail to operate properly, the selected KRA or KRAs will instead perform key recovery as shown in Figure 7.

**Step 1:**$Dev_{A}$ refers to the total number of KRAs *n* and the number of KRAs required for key recovery as mr.**Step 2:**$Dev_{A}$ calculates the number of KRAs *t* required to distribute $TT_{i}$.
(20)t=n−mr**Step 3:**$Dev_{A}$ selects a KRA or KRAs to replace the failed $KRA_{i}$ as follows:
(21)$$\begin{array}{c} j=i-mr \end{array}$$
(22)$$\begin{array}{c} KRA_{i}\to \left\{\begin{array}{c} KRA_{i+1},KRA_{i+2},\ldots ,KRA_{i+t}\;\left(i≦mr\right)\\ KRA_{i+1},\ldots ,KRA_{n},KRA_{1},\ldots ,KRA_{j}\;\left(i>mr\;and\;i\neq n\right)\\ KRA_{1},KRA_{2},\ldots ,KRA_{t}\;\left(i=n\right) \end{array}\right\} \end{array}$$**Step 4:**$Dev_{A}$ distributes $TT_{i}$ to selected KRA or KRAs.**Step 5:** If $KRA_{i}$ fail to operate properly, the selected KRA or KRAs obtain $c_{i},r_{i}$ and $s_{i}$ of failed KRA using the distributed $TT_{i}$ and SGN.
(23)$$\begin{array}{cc} TT_{i}⊕SGN & =(Tc_{i}\|Tr_{i}\|Ts_{i})⊕SGN\\ \; & =(Tc_{i}⊕SGN\|Tr_{i}⊕SGN\|Ts_{i}⊕SGN)\\ \; & =(c_{i},r_{i},s_{i}) \end{array}$$

### 4.7. Session Key Recovery Phase

This phase describes how to recover a key if the $Dev_{B}$ requests key recovery as shown in Figure 8.

**Step 1:** When $Dev_{B}$ requests KRF decryption from Med to recover SK, and sends KRF.**Step 2:** Then Med requests KRF decryption from KRC to recover SK, and sends KRF.**Step 3:** KRC upon receiving a request for KRF decryption, obtains $KRF_{i}$ pieces after KRF decrypt with $sk_{KRC}$.
(24)$$\begin{array}{c} D_{sk_{KRC}}(E_{pk_{KRC}}(KRF_{1}\|\ldots \|KRF_{n}\left\|H\left(SGN\right))\right) \end{array}$$**Step 4:** The obtained $KRF_{i}$ pieces are sent to each M-KRA to request decryption.**Step 5:** The requested M-KRA obtain $c_{i},r_{i},s_{i},SGN,TT_{i}$ values with $sk_{KRA_{i}}$.
(25)$$\begin{array}{c} D_{sk_{KRA_{i}}}(E_{pk_{KRA_{i}}}(c_{i}\|r_{i}\|s_{i}\|SGN\|TT_{i})) \end{array}$$**Step 6:** Among the obtained values, $c_{i},r_{i},s_{i},SGN$ values are encrypted with $pk_{KRC}$ and sends to the KRC.
(26)$$\begin{array}{c} E_{pk_{KRC}}(c_{i}\|r_{i}\|s_{i}\left\|SGN\right) \end{array}$$**Step 7:** KRC compares SGN obtained by decrypting the received ciphertext with $sk_{KRC}$ and H(SGN). If they match, $c_{i},r_{i},s_{i}$ pieces are collected and c,r,s are recovered.**Step 8:** KRC recovers the *k* value using the received ciphertext, public parameters, and recovered c,r,s.
(27)$$\begin{array}{c} k=H({\left(pk_{A}\cdot g^{r}\right)}^{s\cdot sk_{KRC}}\;mod\;p) \end{array}$$**Step 9:** Then, KRC divides *k* by $k_{1},k_{2}$.**Step 10:** KRC recovers the SK using the obtained $k_{1}$ and *c*.
(28)$$\begin{array}{c} D_{k_{1}}\left(C\right)=D_{k_{1}}\left(E_{k_{1}}\left(SK\right)\right)=SK \end{array}$$**Step 11:** KRC compares the calculated $H_{k_{2}}\left(SK\right)$ and *r* values using the obtained k2.**Step 12:** If it matches, KRC sends the recovered SK to Med.**Step 13:** Then, Med sends SK to $Dev_{B}$ and the message is decrypted using the received SK.
(29)$$\begin{array}{c} D_{SK}\left(C\right)=D_{SK}\left(E_{k_{1}}\left(M\right)\right)=M \end{array}$$

## 5. Proposed Scheme-II (Key Recovery System Using Proxy Re-Encryption)

In this section, we propose a proposed scheme-II. This scheme is a scheme recovering the lost and corrupted device’s key. This system was designed based on the scheme of Yang et al. [31]. It consists of a setup phase, a key pair generation phase, a Med enforcement phase, and a session key recovery phase, as shown in Figure 9.

### 5.1. System Parameters

The system parameters used in the proposed scheme-II are as follows:***q*:** Prime number**$H_{1}$:** Hash functions, ${\left\{0,1\right\}}^{*}\times G\to Z_{q}^{*}$**$H_{2}$:** Hash functions, $G\to {\left\{0,1\right\}}^{l_{1}+l_{2}}$ for some bit-length $l_{1},L_{2}\in N$**$H_{3}-H_{5}$:** Hash functions, ${\left\{0,1\right\}}^{*}\to Z_{q}^{*}$**$H_{6}$:** Hash functions, $G\to Z_{q}^{*}$**$Part_{i}$:** System participant *i*, ($Dev_{A},Dev_{B},Med,KRC,KRA\in Part_{i}$)***s*:** Master secret key of KGC, $s\in Z_{q}^{*}$**$sk_{i}$:**$Part_{i}$’s private key, $sk_{i}=\left(d_{i},y_{i},z_{i}\right)$**$pk_{i}$:**$Part_{i}$’s public key, $pk_{i}=\left(X_{i},Y_{i},Z_{i}\right)$**KRF:** Key recovery field, $KRF=(KRF_{1},KRF_{2},KRF_{3},KRF_{4})$

### 5.2. Setup Phase

In this phase, the KGC takes the security parameter $1^{\lambda }$ as an input and generates public parameters.

**Step 1:**KGC selects λ-bit large prime *q* and group G of prime order *q*. In addition, a random generator g∈G is selected.**Step 2:**KGC randomly selects master secret key $s\in Z_{q}^{*}$, and compute $S=g^{s}$.**Step 3:**KGC selects Hash function $H_{1},H_{2},H_{3},H_{4},H_{5},H_{6}$.**Step 4:** The message space M and public parameters $params=(G,l_{1},l_{2},q,g,S,H_{1},$$H_{2},$$H_{3},$$H_{4},H_{5},H_{6})$ are published.

### 5.3. Key Pair Generation Phase

In this phase, $Part_{i}$ receives a partial private key from KGC and uses it to generate full private key $sk_{i}$ and public key $pk_{i}$.

**Step 1:**KGC generates parameters $x_{i}\in Z_{q}^{*}$ for participant $Part_{i}$ through the following operation and sends them to $Part_{i}$ through a secure channel.
(30)$$X_{i}=g^{x_{i}}$$
(31)$$d_{i}=x_{i}+sH_{1}\left(ID_{i},X_{i}\right)\;\mathrm{mod}\;q$$**Step 2:**$Part_{i}$ who receives $X_{i},d_{i}$ from KGC, selects Random numbers $y_{i},z_{i}\in Z_{q}^{*}$ and sets $Part_{i}$’s private key $sk_{i}$.
(32)$$sk_{i}=\left(d_{i},y_{i},z_{i}\right)$$**Step 3:**$Part_{i}$ generates $Y_{i},Z_{i}$ and sets public key $pk_{i}$.
(33)$$Y_{i}=g^{y_{i}}$$
(34)$$Z_{i}=g^{z_{i}}$$
(35)$$pk_{i}=\left(X_{i},Y_{i},Z_{i}\right)$$After that, $Part_{i}$ publishes public key $pk_{i}$.

### 5.4. Session Key Exchange and KRF Generation Phase

In this phase, a session key is exchanged between $Dev_{A}$ and $Dev_{B}$, and a KRF is generated. Furthermore, in the KRF generation phase, after generating KRF, the ciphertext *C* is communicated with KRF as shown in Figure 10.

**Step 1:**$Dev_{A}$ selects $a\in Z_{q}^{*}$ and calcultate partial session key $PSK_{A}$.
(36)$$PSK_{A}=g^{a}$$$Dev_{B}$ also selects $b\in Z_{q}^{*}$ and calculates partial session key $PSK_{B}$.
(37)$$PSK_{B}=g^{b}$$After that, $Dev_{A}$ and $Dev_{B}$ exchange $PSK_{A}$ and $PSK_{B}$ with each other.**Step 2:**$Dev_{A}$ and $Dev_{B}$ calculate the session key $SK={\left(PSK_{B}\right)}^{a}={\left(PSK_{A}\right)}^{b}$ using the exchanged values $PSK_{A}$ and $PSK_{B}$.**Step 3:**$Dev_{A}$ generates the ciphertext message $C=E_{SK}\left(M\right)$ using the generated session key SK.**Step 4:** After that, $Dev_{A}$ selects a random value $t,c\in Z_{q}^{*}$ and $\sigma \in {\left\{0,1\right\}}^{l_{2}}$, and generates KRF using $SK,pk_{Med},pk_{KRC}$ and $pk_{KRA}$ as follows:
(38)$$\pi_{i}=X_{i}\cdot S^{H_{1}\left(ID_{i},X_{i}\right)}$$
(39)$$V_{i}=\pi_{i}^{H_{6}\left(Y_{i}\right)}\cdot Y_{i}$$
(40)$$\tau =H_{5}\;\left(SK,\sigma ,ID_{Med},pk_{Med}\right)$$
(41)$$\alpha =Y_{Med}^{c}$$
(42)$$\beta =Z_{Med}^{c}$$
(43)$$KRF_{1}=g^{t}$$
(44)$$KRF_{2}=Y_{KRC}^{\tau }=g^{\tau \cdot y_{KRC}}$$
(45)$$KRF_{3}=\left(SK\|\sigma \right)⊕H_{2}\left(g^{c}\right)$$
(46)$$KRF_{4}={\left(\alpha \cdot \beta \right)}^{H_{4}\left(V_{KRA}^{\tau }\right)\cdot H_{4}\left(V_{Med}^{\tau }\right)}$$
(47)$$KRF=(KRF_{1},KRF_{2},KRF_{3},KRF_{4})$$After that, $Dev_{A}$ and $Dev_{B}$ communicate with each other using (C‖KRF).

### 5.5. Med Enforcement Phase

In this phase, Med will start recovering the encrypted session key between $Dev_{A}$ and $Dev_{B}$ at the request of $Dev_{A}$ as shown in Figure 11.

**Step 1:**$Dev_{A}$ sends KRF to Med to recover the session key SK.**Step 2:**Med generates the re-encryption key $RK_{Med\to KRC}$.
(48)$$\gamma_{KRC}=X_{KRC}\cdot S^{H_{1}\left(ID_{KRC},X_{KRC}\right)}$$
(49)$$K_{Med-KRC}=H_{3}\left(\gamma_{KRC}^{z_{Med}},Z_{KRC}^{z_{Med}},ID_{Med},pk_{Med},ID_{KRC},pk_{KRC}\right)$$
(50)$$RK_{Med\to KRC}=\left(d_{Med}H_{6}\left(Y_{Med}\right)+y_{Med}\right)\cdot K_{Med-KRC}$$**Step 3:**Med requests key recovery by sending the $\left(KRF_{1},KRF_{2}^{{}^{\prime }},KRF_{4},KRF_{6},RK_{Med\to KRC}\right)$ to the KRC.

### 5.6. Session Key Recovery Phase

In this phase, KRC receives a key recovery request from Med. KRF calculates $KRF_{2}^{\prime }$ using its private key, and then requests key recovery from KRA as shown in Figure 12.

**Step 1:** After receiving $\left(KRF_{1},KRF_{2},KRF_{4},RK_{Med\to KRC}\right)$, KRC calculates $KRF_{2}$ as $KRF_{2}^{{}^{\prime }}$ using its $sk_{KRC}$ as follows:
(51)$$\begin{array}{cc} KRF_{2}^{{}^{\prime }} & =KRF_{2}^{1/y_{KRC}}\\ \; & =Y_{KRC}^{\tau /y_{KRC}}\\ \; & =g^{\tau \cdot y_{KRC}/y_{KRC}}\\ \; & =g^{\tau } \end{array}$$After that, KRC sends the generated $\left(KRF_{1},KRF_{2}^{{}^{\prime }},KRF_{4},RK_{Med\to KRC}\right)$ to the KRA.**Step 2:** After receiving $\left(KRF_{1},KRF_{2}^{{}^{\prime }},KRF_{4},RK_{Med\to KRC}\right)$, KRA re-encrypts $KRF_{2}^{{}^{\prime }}$ as $KRF_{2}^{{}^{\prime \prime }}$ using its $RK_{Med\to KRC}$ as follows:
(52)$$\begin{array}{cc} KRF_{4}^{{}^{\prime }} & =KRF_{4}^{1⁄H_{4}\left(KRF_{2}^{{}^{\prime }d_{KRA}\cdot H_{4}\left(Y_{KRA}\right)+y_{KRA}}\right)}\\ \; & ={\left(\alpha \cdot \beta \right)}^{H_{4}\left(V_{KRA}^{\tau }\right)\cdot H_{4}\left(V_{Med}^{\tau }\right)⁄H_{4}\left(c_{2}^{{}^{\prime }d_{KRA}H_{4}\left(Y_{KRA}\right)+y_{KRA}}\right)}\\ \; & ={\left(\alpha \cdot \beta \right)}^{H_{4}\left(V_{Med}^{\tau }\right)} \end{array}$$
(53)$$KRF_{2}^{{}^{\prime \prime }}={KRF_{2}^{{}^{\prime }}}^{RK_{Med\to KRC}}$$After that, KRA sends $\left(KRF_{2}^{{}^{\prime \prime }},KRF_{4}^{{}^{\prime }}\right)$ to the KRC.**Step 3:**KRC re-decrypts $KRF_{4}^{{}^{\prime }}$ to obtain $KRF_{4}^{{}^{\prime \prime }}$ using $sk_{KRC}$ as follows:
(54)$$K_{Med-KRC}=H_{3}\left(Z_{Med}^{d_{KRC}},Z_{Med}^{z_{KRC}},ID_{Med},pk_{Med},ID_{KRC},pk_{KRC}\right)$$
(55)$$\begin{array}{cc} KRF_{4}^{{}^{\prime \prime }} & ={KRF_{4}^{{}^{\prime }}}^{1⁄H_{4}\left({KRF_{2}^{{}^{\prime \prime }}}^{1⁄K_{Med-KRC}}\right)}\\ \; & ={\left(\alpha \cdot \beta \right)}^{H_{4}\left(V_{Med}^{\tau }\right)⁄H_{4}\left({KRF_{2}^{{}^{\prime \prime }}}^{1⁄K_{Med-KRC}}\right)}\\ \; & =(\alpha \cdot \beta ) \end{array}$$After that, KRC sends $KRF_{4}^{{}^{\prime \prime }}$ to Med.**Step 4:**Med decrypts $KRF_{4}{}^{\prime \prime }$ to obtain SK as follows:
(56)$$\begin{array}{cc} g^{c} & ={KRF_{4}^{{}^{\prime \prime }}}^{1⁄\left(y_{Med}+z_{Med}\right)}\\ \; & ={\left(\alpha \cdot \beta \right)}^{1⁄\left(y_{Med}+z_{Med}\right)}\\ \; & ={\left(Y_{Med}\cdot Z_{Med}\right)}^{1⁄\left(y_{Med}+z_{Med}\right)}\\ \; & =g^{c\cdot (y_{Med}+z_{Med})/\left(y_{Med}+z_{Med}\right)} \end{array}$$
(57)$$\left(SK\|\sigma \right)=KRF_{3}⊕H_{2}\left(g^{c}\right)$$After that, Med sends SK to $Dev_{A}$.**Step 5:**$Dev_{A}$ decrypts the message *M* using the obtained SK.
(58)$$M=Dec_{SK}\left(C\right)$$

## 6. Analysis of the Proposed Schemes

This section explores whether the abovementioned security requirements are satisfied by the two proposed schemes, as shown in Table 1.

### 6.1. Proposed Scheme-I (Key Recovery System Using Signcryption)

KRF integrity: The device, Med, KRA, and KRC participating in key recovery should not be able to transform a device key that generates a KRF maliciously. To solve this problem, this includes the session key hash in parameter *r* of the KRF. Therefore, KRF data cannot be forged. Only the device can access the KRF session key generated by the device.
(59)$$\begin{array}{cc} r & =?\;r^{\prime }\\ H_{k_{2}}\left(g^{x}\;mod\;p,SK\right) & =?\;H_{k_{2}}\left(g^{x}\;mod\;p,SK^{\prime }\right) \end{array}$$Data confidentiality: In the proposed scheme-I, communication between devices is performed through a session key. Therefore, if the session key for the corresponding communication is unknown, the malicious user will not be able to obtain the message. In addition, as the KRF generated in the communication process contains the public keys of KRC and M−KRA, third-party besides KRC and KRA cannot know the contents of the corresponding KRF.Non-repudiation: If the device generates and uses the wrong KRF, KRC cannot recover the key. To solve this problem, the device should not be able to reject the fact that it generated KRF. Therefore, this includes the private key $sk_{A}$ of the device in parameter *s* of the KRF. The device cannot deny that it generated the KRF.
(60)$$\begin{array}{cc} k & ={\left(pk_{A}\cdot g^{r}\right)}^{s\cdot sk_{KRC}}\\ \; & ={\left(g^{sk_{A}}\cdot g^{r}\right)}^{s\cdot sk_{KRC}}\\ \; & ={\left(g^{sk_{A}}\cdot g^{r}\right)}^{x/\left(r+sk_{A}\right)\cdot sk_{KRC}}\\ \; & =g^{sk_{KRC}x}=pk_{KRC}^{x} \end{array}$$Attack on group authentication detection: Malicious key recovery by third-party KRAs should not be possible. Therefore, a lawful KRA group member applies an XOR operation on the values from $R_{KRA_{1}}$ to $R_{KRA_{n}}$ to generate a shared group value of SGN between groups. The device receives it from a lawful group member and hashes the SGN to include H(SGN) in the KRF. When KRC recovers the complete key, it hashes and compares the SGN sent by the M−KRA with the SGN contained in the KRF to ensure it was received from a lawful KRA.
(61)$$\begin{array}{cc} H(SGN) & =?\;H(SGN^{\prime }) \end{array}$$Single point of failure protection: As both the KRC and all KRAs participate in session key recovery, it should be possible to recover the key even if some KRAs fail. Therefore, a special value TT is generated. If some KRAs fail to recover the session key pieces, other KRAs recover the session key pieces instead of the failed KRA and send them to the KRC. TT includes all $c_{i},r_{i},s_{i}$ pieces and the SGN produced by the XOR operation. The other KRA (not the corresponding KRA) decrypts $TT_{i}$ and sends it to the KRC, allowing the KRC to recover the complete session key.
(62)$$\begin{array}{cc} TT_{i} & =(Tc_{i}\|Tr_{i}\|Ts_{i}) \end{array}$$
(63)$$\begin{array}{cc} TT_{i}⊕SGN & =(\left(c_{i}⊕SGN\right)\|\left(r_{i}⊕SGN\right)\|\left(s_{i}⊕SGN\right))\\ \; & =(\left(c_{i}⊕SGN\right)⊕SGN\|\left(r_{i}⊕SGN\right)⊕SGN\|\left(s_{i}⊕SGN\right)⊕SGN)\\ \; & =(c_{i}\|r_{i}\|s_{i}) \end{array}$$Med applied for support: Med should be able to view the encrypted data by acquiring the encrypted session key in the event of an emergency where it is necessary to view the device’s data. Therefore, Med sends KRF to KRC, and KRC decrypts KRF to obtain KRF pieces. The acquired KRF pieces are sent to M−KRA and requested for recovery. Then, M−KRA obtains session key pieces by decrypting the acquired KRF pieces. The obtained session key pieces are sent to KRC, and KRC recovers the complete session key. After that, it sends the complete session key to the Med, allowing message decryption.Key escrow problem: The proposed scheme-I is based on a CL-PKC scheme. Therefore, as KGC can generate only a part of the private key during the private key generation process, the key escrow problem caused by KGC in ID-based encryption has been solved.

### 6.2. Proposed Scheme-II (Key Recovery System Using Proxy Re-Encryption)

KRF integrity: In this proposed scheme-II, KRF is encrypted with the public keys of Med and KRA. Therefore, during the key recovery process, Med, KRC and KRA cannot be forged or modified KRF by alone.
(64)$$KRF=(KRF_{1},KRF_{2},KRF_{3},KRF_{4})$$
(65)$$\pi_{Med}=X_{Med}\cdot S^{H_{1}\left(ID_{Med},X_{Med}\right)}$$
(66)$$V_{Med}=\pi_{Med}^{H_{6}\left(Y_{Med}\right)}\cdot Y_{Med}$$
(67)$$KRF_{4}={\left(\alpha \cdot \beta \right)}^{H_{4}\left(V_{KRA}^{\tau }\right)\cdot H_{4}\left(V_{Med}^{\tau }\right)}$$Data confidentiality: As the KRF generated in the communication process contains the public key of the Med and the secret values of KRC and KRA, third-party besides the Med, KRC, and KRA cannot know the contents of the corresponding KRF. In addition, even if all three of the Med, KRC, and KRA do not participate, each Med, KRC, and KRA cannot know the contents of the KRF.Med applied for support: Med can perform recovery of SK as needed. KRF is created using the public key of Med. Med can perform the key recovery process when it is determined that the key recovery is necessary for Dev that it manages. For this, Med can create $RK_{Med\to KRC}$ and request and execute the key recovery process through KRC and KRA.Collusion attack resistance: Fewer than three participants among the Med, KRC, and KRA must be prevented from maliciously acting together, thus preventing recovery of the key, and unauthorized entities must be prevented from obtaining the key. Therefore, the Med requires the cooperation of the KRC and KRA to decrypt KRF. Thus, even if the Med has colluded with a single participant among the KRC and KRA, the completed key recovery cannot be achieved without the assistance of the third participant as follows:
(68)$$KRF=(KRF_{1},KRF_{2},KRF_{3},KRF_{4})$$
(69)$$KRF_{3}=\left(SK\|\sigma \right)⊕H_{2}\left(g^{c}\right)$$In order to obtain SK from the above $KRF=(KRF_{1},KRF_{2},KRF_{3},KRF_{4})$, $KRF_{3}$ must be decrypted. In order to decrypt $KRF_{3}$, Med,KRC and KRA need to know *c* or $g^{c}$. However, *c* and $g^{c}$ know only Dev. Therefore, it is necessary to obtain $g^{c}$ by decrypting $KRF_{4}$.
(70)$$KRF_{4}={\left(\alpha \cdot \beta \right)}^{H_{4}\left(V_{KRA}^{\tau }\right)\cdot H_{4}\left(V_{Med}^{\tau }\right)}$$Here, $KRF_{4}$ contains $\alpha \cdot \beta =Y_{Med}^{c}\cdot Z_{Med}^{c}$, so the attackers are $H_{4}\left(V_{KRA}^{\tau }\right)$ and $H_{4}\left(V_{Med}^{\tau }\right)$ should be computed.
(71)$$\pi_{i}=X_{i}\cdot S^{H_{1}\left(ID_{i},X_{i}\right)}$$
(72)$$V_{i}=\pi_{i}^{H_{6}\left(Y_{i}\right)}\cdot Y_{i}$$Since $V_{i}$ can be created using a public key, anyone can create it. However, since τ only knows Dev, attackers must use $KRF_{2}$ to calculate $H_{4}\left(V_{KRA}^{\tau }\right)$ and $H_{4}\left(V_{Med}^{\tau }\right)$.
(73)$$KRF_{2}=Y_{KRC}^{\tau }=g^{\tau \cdot y_{KRC}}$$Here, a KRC’s private key $y_{KRC}$ is required to obtain $g^{\tau }$ from $KRF_{2}$. Therefore, KRC is required in the key recovery process.
(74)$$KRF_{2}^{\prime }=g^{\tau }=KRF_{2}^{1/y_{KRC}}=g^{\tau \cdot y_{KRC}/y_{KRC}}$$Next, since the attacker does not know τ, he has to perform the following operation to calculate $H_{4}\left(V_{KRA}^{\tau }\right)$. In the end, the KRA’s private keys $d_{K}RA$ and $y_{K}RA$ are required, so KRA is also required.
(75)$$\begin{array}{cc} KRF_{4}^{{}^{\prime }} & =KRF_{4}^{1⁄H_{4}\left(KRF_{2}^{{}^{\prime }d_{KRA}\cdot H_{4}\left(Y_{KRA}\right)+y_{KRA}}\right)}\\ \; & ={\left(\alpha \cdot \beta \right)}^{H_{4}\left(V_{KRA}^{\tau }\right)\cdot H_{4}\left(V_{Med}^{\tau }\right)⁄H_{4}\left(c_{2}^{{}^{\prime }d_{KRA}H_{4}\left(Y_{KRA}\right)+y_{KRA}}\right)}\\ \; & ={\left(\alpha \cdot \beta \right)}^{H_{4}\left(V_{Med}^{\tau }\right)} \end{array}$$Furthermore, an attacker who acquires (α·β) must compute $g^{c}$ to obtain (SK‖σ) from $KRF_{3}=\left(SK\|\sigma \right)⊕H_{2}\left(g^{c}\right)$.
(76)$$\begin{array}{cc} g^{c} & ={\left(\alpha \cdot \beta \right)}^{1⁄\left(y_{Med}+z_{Med}\right)}\\ \; & ={\left(Y_{Med}\cdot Z_{Med}\right)}^{1⁄\left(y_{Med}+z_{Med}\right)}\\ \; & ={\left(g^{c\cdot y_{Med}}\cdot g^{c\cdot z_{Med}}\right)}^{1⁄\left(y_{Med}+z_{Med}\right)}\\ \; & =g^{c\cdot (y_{Med}+z_{Med})/\left(y_{Med}+z_{Med}\right)} \end{array}$$In order to acquire $g^{c}$ using $KRF_{4}^{{}^{\prime }}$, Med’s private keys $y_{Med}$ and $z_{Med}$ are required, so Med is also required. As a result, in order to obtain SK by decrypting KRF, all of Med,KRC, and KRA must participate.Key escrow problem: The proposed scheme-II is based on a CL-PKC scheme. Therefore, as KGC can generate only a part of the private key during the private key generation process, the key escrow problem caused by KGC in ID-based encryption has been solved.

## 7. Conclusions

This paper proposed key recovery systems based on key encapsulation secured from various attacks in IoMT environments in schemes II and II.

In the key recovery system, the session key used in the ciphertext is recovered via the KRF and used. However, the KRF can be forged and KRF owners can deny the fact that they generated the KRF. Furthermore, unauthorized KRAs can access the M-KRA and interfere with key recovery. To solve this problem, the key recovery system using signcryption includes the session key hash in the KRF. Therefore, the KRF data cannot be forged. In addition, this system includes the private key of a device in special value of the KRF. A device cannot deny that it generated the KRF. Furthermore, the system ensures the security requirements mentioned in Section 3, including KRF integrity and non-repudiation, are fulfilled.

Additionally, there is a problem that the key can be recovered by collusion attacks and key or message leakage among the Med, KRC, and KRA. To solve this problem, the Med must have the help of the KRC and KRA to recover the key by a proxy re-encryption function. In addition, the KRC or KRA would also need mutual help to recover a complete session key. That is, by limiting the information and processing capabilities of the three participants, the key recovery system can be expected to be secure against various attacks. Furthermore, because the KGC generates the private keys of all participants, there is the problem that the KGC’s authority is strong. To solve this, a partial private key generation scheme is used. The KGC generates a partial private key and sends it to the participants. Participants who receive partial private keys use them to generate complete private keys and solve the KGC key escrow problem.

Future research is to check whether unexpected problems occur when the proposed schemes are implemented in actual systems. Furthermore, additional research is needed that can examine the amount of computations, time, and cost incurred when recovering keys. In addition, further research is needed to determine whether the proposed schemes are secure against other types of security threats.

## Figures and Tables

**Figure 1 sensors-21-03474-f001:**
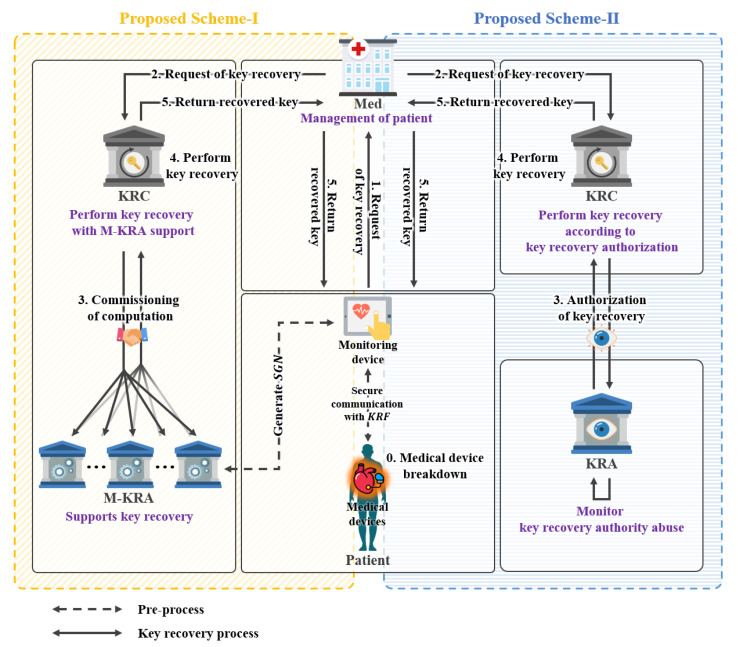
Summary and comparison of proposed key recovery system model.

**Figure 2 sensors-21-03474-f002:**
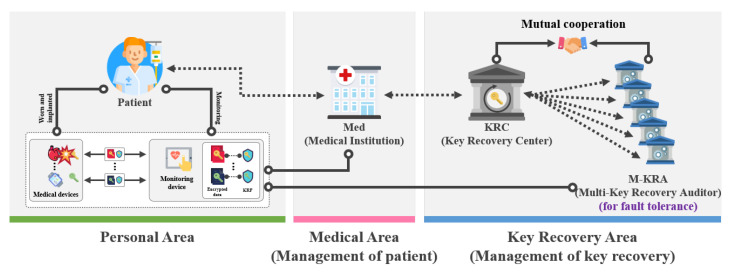
Proposed scheme-I.

**Figure 3 sensors-21-03474-f003:**
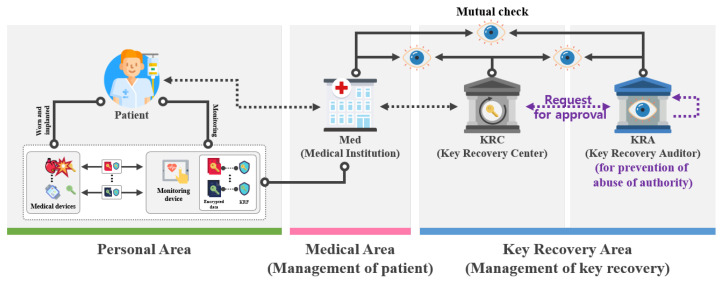
Proposed scheme-II.

**Figure 4 sensors-21-03474-f004:**
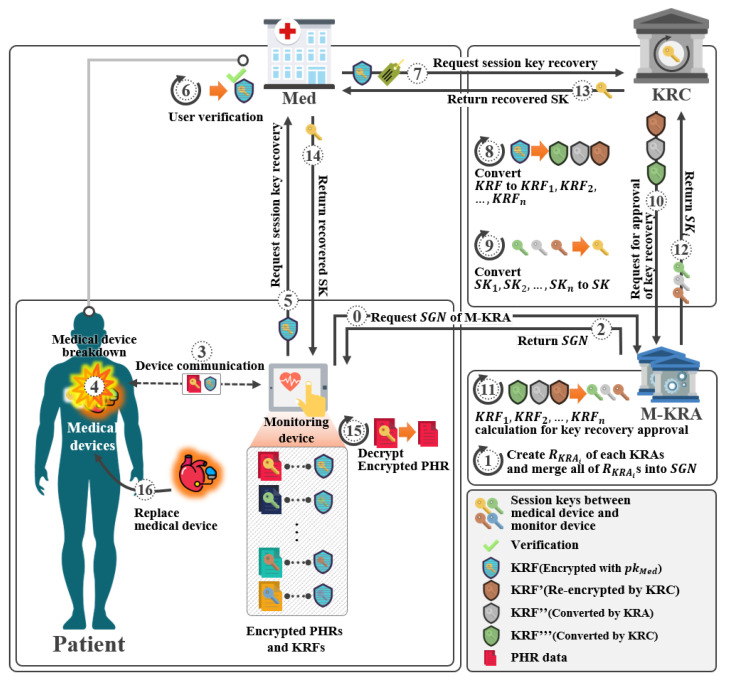
Scenario of proposed scheme-I.

**Figure 5 sensors-21-03474-f005:**
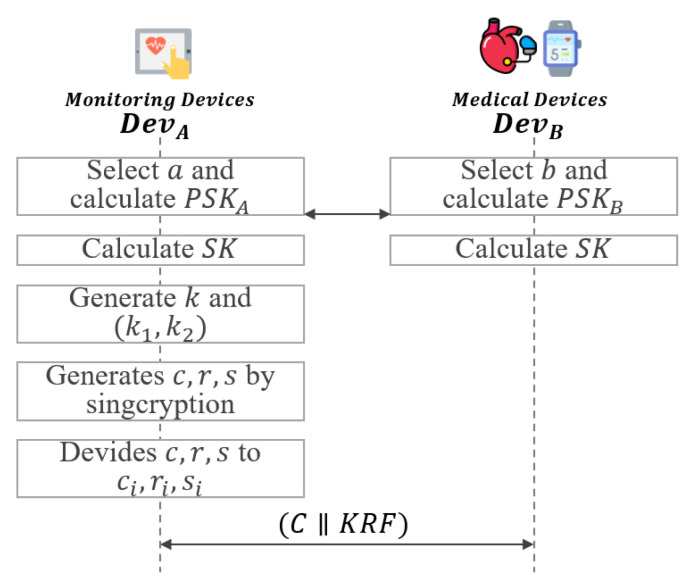
Session key exchange and encryption phase of proposed scheme-I.

**Figure 6 sensors-21-03474-f006:**
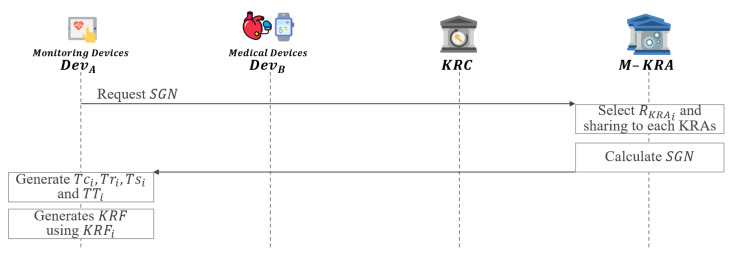
KRF generation phase of proposed scheme-I.

**Figure 7 sensors-21-03474-f007:**
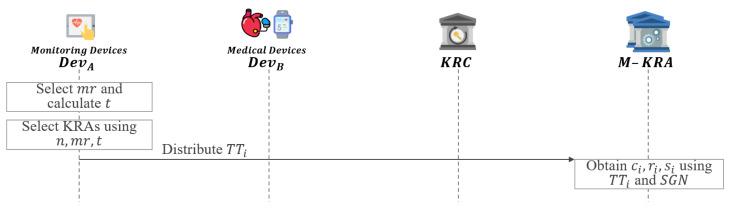
KRA fault recovery phase of proposed scheme-I.

**Figure 8 sensors-21-03474-f008:**
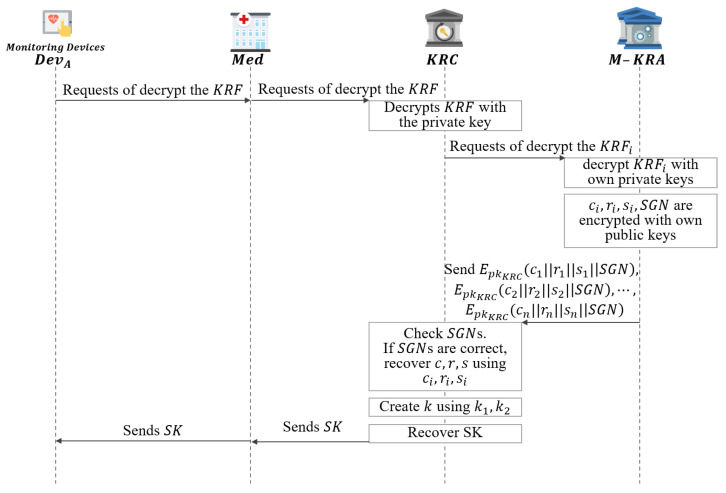
Session key recovery phase of proposed scheme-I.

**Figure 9 sensors-21-03474-f009:**
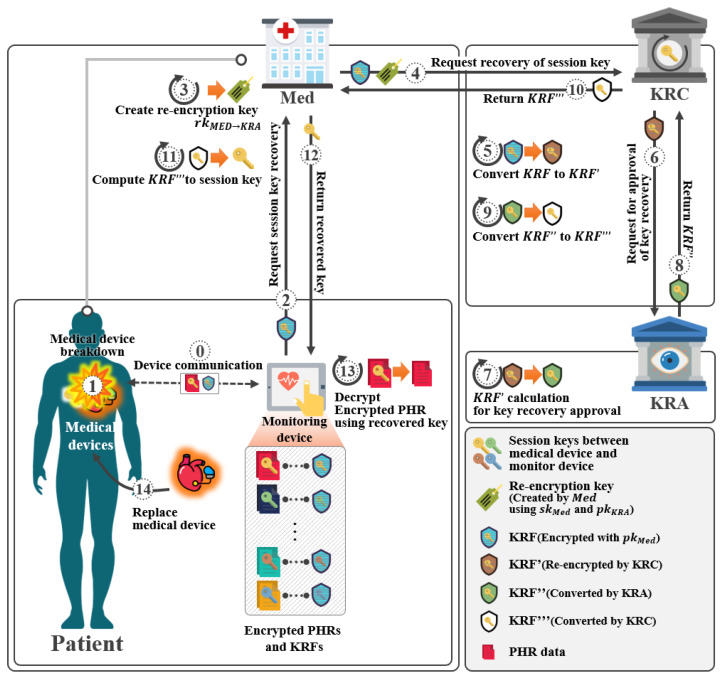
Scenario of proposed scheme-II.

**Figure 10 sensors-21-03474-f010:**
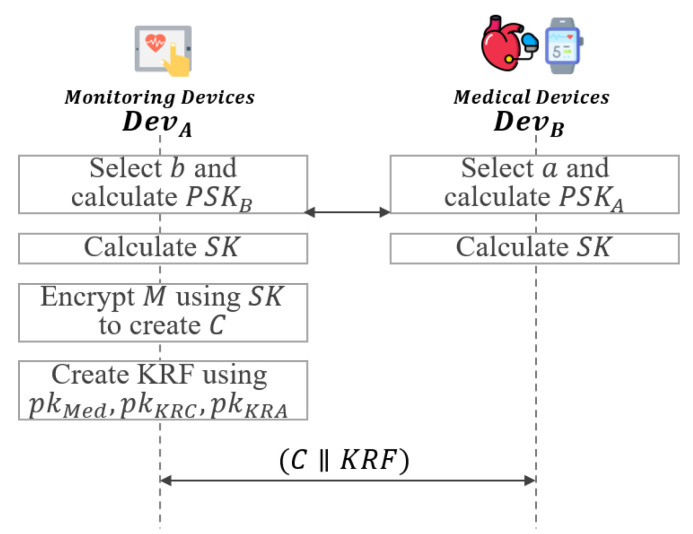
Session key exchange and KRF generation phase of proposed scheme-II.

**Figure 11 sensors-21-03474-f011:**
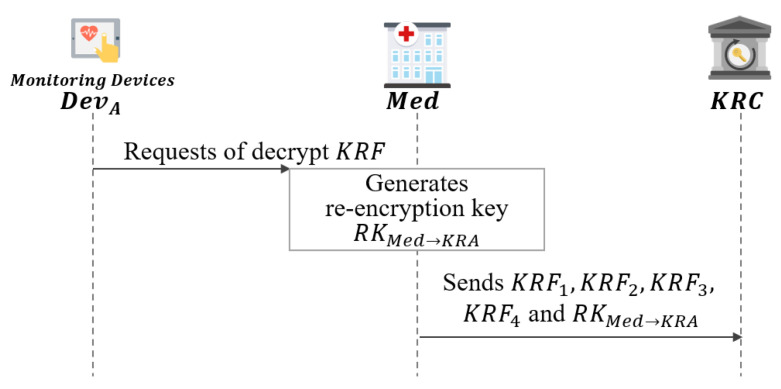
Med enforcement phase of proposed scheme-II.

**Figure 12 sensors-21-03474-f012:**
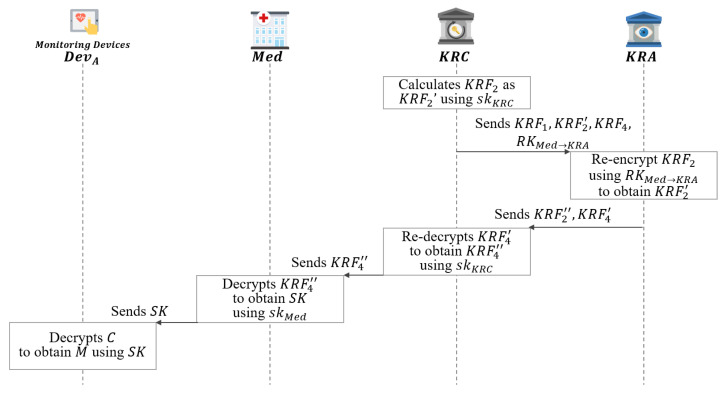
Session key recovery phase of proposed scheme-II.

**Table 1 sensors-21-03474-t001:** Comparison of proposed schemes.

	[6]	[7]	[8]	[10]	[11]	[12]	[13]	Proposed Scheme-I	Proposed Scheme-II
KRF integrity	-	-	-	×	×	×	×	∨	∨
Non-repudiation	-	-	-	×	×	×	×	∨	×
Attack on group authentication detection	-	∨	-	∨	×	×	∨	∨	-
Single point of failure protection	×	×	×	∨	∨	×	×	∨	×
Data confidentiality	∨	∨	∨	∨	∨	∨	∨	∨	∨
Med applied for support	-	-	-	-	-	-	-	∨	∨
Collusion attacks resistance	∨	×	×	×	×	×	×	×	∨
Key escrow problem	×	×	×	∨	×	×	×	∨	∨
∨: Provided/×: Not provided/-:Not considered									

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
