# Peer review of "Secure Encapsulation Schemes Using Key Recovery System in IoMT Environments"

_sensors, 2021, doi:10.3390/s21103474_

Round 1

Reviewer 1 Report

In this paper, the authors proposed a secure encapsulation scheme for key recovery. However, lots of parameters are not properly defined especially in Section 4. The English should be double-checked by native speakers.

Problems in System Proposition

1) page 8, line 308, it seems that $j_j$ is not explicitly used in the system

2) page 8, line 313, is SGN open or close? It seems that SGN should be closed. Otherwise, the proposed scheme is not secure. It also should be clearly presented in the paper.

3) page 8, line 315-317, $c_i$, $r_i$, and $s_i$ are not defined. For clear understanding, the parameters should be defined before using in the scheme.

4) page 8, line 320, it seems that $TT_i$ is only XORed with SGN. Is there any reason to generate $Tci, Tr_i, Ts_i$ individually XORed with SGN if you only use $TT_i$ in the scheme?

5) page 8, line 322, who owns the public key $PK_{ag_i}$? Not defined.

6) page 9, line 344, here, $c$, $r$, and $s$ are GENERATED. However, $s$ is already generated as a private key. Is it not the same as the private key $s$, or what? Also, the exact way of generating these parameters is not specified. It is important because $k$ is recovered at the session key recovery phase using these parameters. However, if you don't know how to generate $c$, $r$, $s$ from $k$, then how can you recover $k$ from them?

7) page 9, line 351, Is SGN transmitted via a secure channel? It should be specified. (see 2))

8) page 9, line 372, $H_{k_2} (SK)$ is not the same as defined hash functions, $H_1$, $H_2$, or $H_3$.

9) page 9, line 390, it seems that $j_j$ is not used again

10) page 10, line 394, $g_KRC$ is not defined. Is it the same as $g$ or not?

11) page 10, line 395, What is $Z_KRA$? It is also used without any definition.

12) page 10, line 395, $r$ is not defined.

13) page 10, line 395, $Y^T_A$ is not defined.

14) page 10, line 398, $T_{KRC->Med}$ is not defined. What is $1/T_{KRC-> Med}$ in the exponent? Does it mean the $T_{KRC->Med}$-th root of unity or what?

15) page 10, line 418, The generation method of $A_i$ and $B_i$ is not specified. Does it mean $A_i = g^{a_i}$?

16) page 11, line 422, is session key $SK$ randomly generation here?

17) Finally, on page 4, line 160, the authors said that scheme 2 is designed using the bilinear map. However, in Section 4, I cannot find the exact use of the bilinear map in the proposed scheme 2. There is no explicit use of the bilinear map.

Misc...

1) page 2, line 45, "when the key used in the ciphertext is needed..." Please, rephrase it since it is hard to figure out the exact meaning.

2) page 3, line 101, please define "LEA" before using it.

Author Response

Dear Reviewer 1,

The reviewer gave me good feedback and became a higher-quality paper.

It was a very rewarding experience for me.

Thank you again for your hard work, it is greatly appreciated.

Best Regards

Teahoon Kim

Reviewer 2 Report

The paper is covering in detail the very important topic of secret key recovery. This paper is unfortunately extremely poorly written, poor english, repetitive, hard to read.

I strongly recommend the authors to profoundly review/re-write this paper, therefore I am providing some recommendations that will be hopefully helpful.

On a positive standpoint, the methods suggested by the authors for key recovery are well thought. The paper is based on solid engineering ground.

Here a list of recommendations to improve this paper:

1-Abstract/Introduction

Please eliminate the generalities from the abstract, erase most of lines 1 to 13. An abstract should provide a summary of the paper, not background informations such as the fourth industrial revolution. Avoid the use of acronyms in thee abstract.

You need to restructure lines 72 to 91. You discuss scheme 1 lines 72 to 75, then scheme 2 lines 76 to 79, then you go back to discussing scheme 1. One way to improve this is to move down lines 76 to 79. Do not use the word paper twice in a row like you are doing lines 80 and 81.

2- Review the english, and avoid translating from your native language into english. Here an example:

Line 65-66: In this way, it is said that safety is necessary for key management and recovery in various environments using IoT. 

The first section of the sentence should be removed, and directly start with: "Safety is necessary for...."

You could ask native speaker to review your documents.

3- typos, and over-complicated symbols.

For example I see typos lines 153, 157, 398, 

4- The figures 1 to 4 need to be totally rethough. Please keep in mind that this is a journal without limitation in the length. You probably need to generate twice as much quality figures to really explain your work.

Fig. 1 and Fig 2 are almost the same. You should not copy paste Fig. 1 to generate Fig.2, then make little changes. We have to look at the small details between identical blocks to see the difference. One option is to generate a generic figure describing both Fig 1 and Fig 2, then explain separately in additional figures the differences.

Same comment on Fig. 3 and 4, which are almost identical. In addition these figures are not readable. You are providing too much non-readable details. Most of your readers will have difficult time to see the differences between the figures, and lose interest.

5- Repetitions between scheme 1 (sections 3.1 and 4.1) and scheme 2 (sections 3.2 and 4.2) make the paper hard to read. As discussed in the figures, you need to start by describing the common points between the two, and the differences. Copy paste of 90% of the common scheme should be avoided.

Your readers will not be interested to read about the same information twice, which they will be interested to understand the differences, and respective values.

6- In tables 1 and 2, you again copy pasted table 1 to get table 2. You should combine scheme 1 and 2 in the same table, and discuss respective values. 

7- Conclusion. Your should again eliminate the generalities. Lines 517-530 should be removed. You should add more informations on futures plans and studies.

8- The reference section is weak. Over the 17 references, 7 are more than 20 years old, and only 6 are recent (5 years). This is too light, and should be completed with a description of recent work.

Author Response

Dear Reviewer 2,

The reviewer gave me good feedback and became a higher-quality paper.

It was a very rewarding experience for me.

Thank you again for your hard work, it is greatly appreciated.

Best Regards

Teahoon Kim

Reviewer 3 Report

This paper deals with the relevant issue of cryptography public key recovery in distributed systems, such as medical IoT. Personal devices generate health data to be transmitted to medical centers. If these devices´ private keys are lost or compromised, the need for key recovery is paramount, thus requiring a scheme to reinforce a legitimate key recovery process.

Although the paper presents good arguments for its proposed key recovery framework, some issues still need clarifications and additional content, as pointed out below.

The paper needs a broad revision of English grammar, beginning in the title "A Secure Encapsulation schemes..." should be corrected to "A Secure Encapsulation schemes base" for the sake of good English concordance. Similar corrections are needed all over the text.

For a better organization, it seems that section "3.1.1. System Configuration" and Figure 1 should be moved to the beginning of Section 2 since the presentation of the concepts would benefit from a  general presentation of key recovery frameworks.

The Introduction must not refer to the special values y and z (line 91) since the paper will provide their definition afterward.

The acronyms ITS, LEA, M-KRA, KGC, CDH, are used but not expelled the first time they appear.

Please define what a "Traditional encryption scheme" (line 144) is.

Section "2.6. Proxy Re-Encryption" must be rewritten to clarify the roles of A and B and Alice and Bob.

A sequence diagram is suggested to improve the protocol's presentation summarized in lines 208-215. 

Figure 2 is almost the same as Figure 1, and the lines 238-261 are too similar to lines 185-206, so this repetition seems unnecessary, or the authors must state at least the differences.

No Figure 2.b is referred to in line 264.

It is not clear how the Med gets the passphrases and KRF of malicious actors, as stated in line 265.

A sequence diagram is suggested to improve the protocol's presentation, summarized in lines 263-271. 

Regarding the expression "two or fewer" in line 284, the authors must review the whole text since "fewer than two" is just one, which is obvious.

A sequence diagram is suggested to improve the protocol's presentation shown in Figure 3, with the steps more clearly disposed of as a sequence of events and messages involving multiple parts. 

Although Section 5 provides an informal security analysis of the proposed scheme, it is important to extend the analysis with formal proof regarding the proposed security protocol, either analytical proof (See for instance: https://doi.org/10.3390/s18092813) or the verifications provided by a security protocol validation tool. 

Author Response

Dear Reviewer 3,

The reviewer gave me good feedback and became a higher-quality paper.

It was a very rewarding experience for me.

Thank you again for your hard work, it is greatly appreciated.

Best Regards

Teahoon Kim

Round 2

Reviewer 2 Report

It is an excellent research work, however it is hard to read in its current form.

1- The authors are presenting too much details that make the paper extremely hard to read. You should provide some of the information as attachment, allowing the reader to go through the paper more rapidly, then digging for details as needed. For example a simplified version of section 4.1 (lines 323 to 411) has to be re-written.

2- There is no need to repeat the same information on section 4.2. Please present only the differences.

3- The Figure 4, 6, and 7 are excellent. However Figures 3 and 8 are too complex, and almost the same. The 2/3 of the figures on the left are exactly the same. You should instead present one figure showing the common diagram, then two separate pictures detailing what is on the left. Same point on Fig 1 and 2.

4- You also have to review your document and fix some small issues. For example lines 198-200 is almost a copy paste of lines 196-198. You will lose readers with all the repeats across the document.

Author Response

Cover letter

April 28, 2021

Department of Software Convergence,

Soonchunhyang University

22,Soonchunhyang-ro, Asan-si, Republic of Korea

Dear Editor

I am submitting a manuscript for publication in Sensors Journal. The manuscript is entitled “Secure Encapsulation Schemes using Key Recovery System in IoMT Environments”. It has not been published elsewhere and has not been submitted simultaneously for publication elsewhere. The evolution to a hyper-intelligent and hyper-connected society using IoT from the 4th industrial revolution is attractive. However, in an IoT-enabled environment, key management is very important, and if the key is lost or corrupted, the key must be recovered. In particular, key management is more important in IoT environments related to human life such as IoMT. To solve the above problem, we propose safe key recovery techniques in the IoMT environment. These proposed methods can safely recover keys from various security threats. Through this paper, we look forward to safe key recovery methods in the IoMT environment.

Thank you very much for your consideration. Yours Sincerely,

Ph.D Course Taehoon Kim

Soonchunhyang University

22,Soonchunhyang-ro, Asan-si, Republic of Korea

Tel.: +82-10-6484-4543

E-mail: 20134101@sch.ac.kr

Ph.D Course Wonbin Kim

Soonchunhyang University

22,Soonchunhyang-ro, Asan-si, Republic of Korea

Tel.: +82-10-9948-8988

E-mail: wbkim29@sch.ac.kr

M.S. Course Daehee Seo

Sangmyung University

7-1, Jongno-gu, Seoul, Republic of Korea

Tel.: +82-10-6733-3227

E-mail: daehseo@smu.ac.kr

Corresponding author: Im-Yeong Lee

Soonchunhyang University

22,Soonchunhyang-ro, Asan-si, Republic of Korea

Tel.: +82-10-5405-7038

E-mail: imylee@sch.ac.kr

Reviewer 3 Report

Although the paper presents improvements based on comments from the first review round, it still needs corrections and clarifications in this second submitted version.

I have appreciated that the authors accepted the suggestions regarding the organization, style, corrections of figures, definitions of variables and concepts, and the clarification of acronyms. Also, it is interesting to see the suggested sequence diagrams to improve the protocol's presentation. 

Notwithstanding, the paper still presents many English grammar issues, beginning in the title that I suggest to be corrected to "A Cryptography Key Recovery System for the IoMT Environment." 

Numerous other corrections are needed all over the text, and I suggest a review by a native English-speaking person. For instance, these are corrections that apply just to the abstract: 

"Recently, while Internet of Things systems have been introduced to facilitate diagnosis and treatment in the healthcare and medical environment, there are many issues concerning threats to these systems' security. For instance, if a key used for encryption is lost or corrupted, then ciphertexts produced with this key cannot be decrypted anymore. Hence, this paper presents schemes for a key recovery system that can recover the lost or the corrupted key on the Internet of Medical Things. In our proposal, when the key used in the ciphertext is needed, this key is obtained from a Key Recovery Field present in the cyphertext. Thus, the recovered key will allow decrypting the ciphertext. However, there are threats to this proposal, including the case of the Key Recovery Field being forged or altered by a malicious user and the possibility of collusion among participating entities (Medical institution, Key Recovery Auditor, and Key Recovery Center) which can interpret the Key Recovery Field and abuse their authority to gain access to the data. To prevent these threats, two schemes are proposed, the first one to enhance the security of a multi-agent key recovery system by providing the Key Recovery Field with efficient integrity and non-repudiation functions, and the second one providing a proxy re-encryption function resistant to the collusion attack against the key recovery system."

Before line 69 in the introduction, it is necessary to outline a key recovery system and define the Key Recovery Auditor (KRA) and the Medical institution (Med) roles. These concepts are used to announce the contributions in lines 70 to 89. The term KGC used in line 88 was not defined previously.

These writing issues really difficult the understanding of the proposals by the reader and raise doubts about the correctness of the proposals and their effectiveness. 

As the protocol operations in the two proposed schemes are rather complex, the paper still lacks formal proof for its proposals. Maybe, the paper should present the possible attacks against the proposed schemes and how the schemes resist these attacks.

Author Response

(The authors gave the same response as above.)
